# Assessment of Iodine and Selenium Nutritional Status in Women of Reproductive Age in Latvia

**DOI:** 10.3390/medicina57111211

**Published:** 2021-11-05

**Authors:** Vija Veisa, Ieva Kalere, Tatjana Zake, Ieva Strele, Marina Makrecka-Kuka, Sabine Upmale-Engela, Andrejs Skesters, Dace Rezeberga, Aivars Lejnieks, Iveta Pudule, Daiga Grinberga, Biruta Velika, Maija Dambrova, Ilze Konrade

**Affiliations:** 1Department of Obstetrics and Gynecology, Riga Stradins University, LV-1007 Riga, Latvia; dace.rezeberga@rsu.lv; 2Department of Internal Disease, Riga Stradins University, LV-1007 Riga, Latvia; Ieva.Kalere@gmail.com (I.K.); Tatjana.Zake@inbox.lv (T.Z.); upsabine@gmail.com (S.U.-E.); aivars.lejnieks@rsu.lv (A.L.); maija.dambrova@rsu.lv (M.D.); ilze.konrade@rsu.lv (I.K.); 3Institute of Occupational Safety and Environmental Health, Riga Stradins University, LV-1007 Riga, Latvia; ieva.strele@rsu.lv; 4Latvian Institute of Organic Synthesis, LV-1006 Riga, Latvia; makrecka@biomed.lu.lv; 5Riga East University Hospital, LV-1038 Riga, Latvia; 6Scientific Laboratory of Biochemistry, Riga Stradins University, LV-1007 Riga, Latvia; andrejs.skesters@rsu.lv; 7Centre for Disease Prevention and Control, LV-1005 Riga, Latvia; iveta.pudule@spkc.lv (I.P.); Daiga.grinberga@spkc.gov.lv (D.G.); Biruta.Velika@spkc.gov.lv (B.V.)

**Keywords:** iodine, urinary iodine concentration (UIC), thyroid, pregnancy, autoimmunity, selenium

## Abstract

*Background and Objectives:* Adequate dietary intake of iodine and selenium is essential during pregnancy. While iodine is vital for maternal thyroid function and fetal development, selenium contributes to the regulation of thyroid function and thyroid autoimmunity. This study aimed to assess the consumption of iodine- and selenium-containing products by women of reproductive age and the iodine and selenium nutritional status of pregnant women in Latvia. *Materials and Methods:* Population health survey (2010–2018) data were used to characterize dietary habits in women of reproductive age. Additionally, 129 pregnant women in the first trimester were recruited; they completed a questionnaire and were tested for thyroid function, urinary iodine concentration (UIC), and serum selenium and selenoprotein P levels. *Results:* The use of some dietary sources of iodine (e.g., milk and dairy products) and selenium (e.g., bread) has decreased in recent years. Less than 10% of respondents reported the use of iodized salt. The use of supplements has become more common (reported by almost 50% of respondents in 2018). Dietary habits were similar in pregnant women, but the use of supplements was even higher (almost 70%). Nevertheless, most supplements used in pregnancy had insufficient contents of iodine and selenium. Thyroid function was euthyreotic in all women, but 13.9% of participants had a thyroid peroxidase antibodies (TPO-ab) level above 60 IU/mL. The median UIC (IQR) was 147.2 (90.0–248.1) μg/gCr, and 52.8% of pregnant women had a UIC below 150 μg/gCr. The mean selenium (SD) level was 101.5 (35.6) μg/L; 30.1% of women had a selenium level below 80 μg/L. The median selenoprotein P level was 6.9 (3.1–9.0) mg/L. *Conclusions:* Iodine nutrition in Latvian population of pregnant women was near the lower limit of adequate and a third of the population had a selenium deficiency. Supplements were frequently used, but most did not contain the recommended amounts of iodine and selenium.

## 1. Introduction

The function of the thyroid is to generate sufficient quantities of two iodine-containing hormones, triiodothyronine (T3) and thyroxine (T4), that are necessary to meet the demands of peripheral tissues. Although thyroid hormones are needed throughout the entire human lifespan to ensure metabolic, developmental, and thermal homeostasis regulation [1], they are the most important during early pregnancy. In recent years, it has been demonstrated that even mild maternal hypothyroxinemia can cause permanent and irreversible disturbances of cerebral cortex cytoarchitecture in the fetus [2], leading to cognitive impairment later in life [3]. Endocrine disruptors or autoimmune thyroid disease can also cause insufficient availability of maternal T4 for neurodevelopment [3,4], but the most important cause is inadequate intake of iodine, an essential substrate for thyroid hormone synthesis. Pregnancy affects almost all aspects of thyroid hormone metabolism, and the estimated increase in T4 production demand in early pregnancy is 20–40% [5]; accordingly, the required iodine intake increases from 150 µg/d to 250 µg/d [6].

As a trace element, iodine can be found in soil and water, and humans obtain it from the diet. The most important dietary source of iodine in human nutrition is traditionally food of animal origin, e.g., milk and fish. Notably, plants do not contain sufficient levels of this element, and therefore the consumption of vegetables cannot ensure adequate dietary iodine intake. Even in iodine-sufficient areas, vegetarians and vegans are susceptible to iodine deficiency [7]. Therefore, various international programs address iodine nutrition to prevent iodine deficiency. These programs range from foodstuff iodation to raising awareness. Prevention of iodine deficiency before conception is more important because recent systematic reviews do not provide an unequivocal inference on the benefits and harms of iodine supplementation during pregnancy. Efforts have been made in Latvia to educate the public on iodine nutrition, which might partially explain the improvement that we found in iodine intake in Latvian schoolchildren from 2010–2011 compared to 2000 [8]. National recommendations on iodine nutrition were proposed [9], but they have not yet been implemented. Considering that in marginally deficient regions such as Latvia, iodine levels decrease during pregnancy [10,11], iodine deficiency needs to be corrected in the earliest stages of pregnancy, preferably before conception.

Selenium in the form of selenocysteine is present in specific selenoproteins, 25 of which have been identified in humans. These selenoproteins are essential for protection against oxidative stress and play a key role in immune system regulation as well as the activation and inactivation of thyroid hormones [12]. The highest concentration of selenium is found in the thyroid gland due to high levels of selenoproteins. Furthermore, selenium is important for the normal development and growth of organisms and has been implicated in adverse health conditions during pregnancy, such as preeclampsia, autoimmune thyroid disease, miscarriage, and preterm birth [13]. In addition, selenium progressively accumulates in the kidneys and heart of the growing fetus as pregnancy continues [14].

The biomarkers of selenium status and function include GPX3 (10–25% of plasma selenium) and SEPP1, which comprise 40–60% of total plasma selenium [15]. SEPP plays a central role in whole body selenium transport from the liver to other tissues and is, therefore, a good indicator of selenium status. Moreover, it exhibits antioxidant anticarcinogenic properties and is implicated in trans-placental transport of selenium to the fetus [16].

Maternal selenium requirements also increase during gestation and lactation, reaching 70 μg/day for pregnant women [17]. Studies show that intake of this amount of selenium is sufficient for the effective functioning of selenoenzymes, but the levels necessary for autoimmunity regulation have not yet been determined [18]. The selenium level in the blood is dependent primarily on recent dietary intake and the bioavailability of the form of selenium taken up. However, both weak [19,20] and strong correlations [21] between dietary selenium intake and plasma/serum, whole blood and toenail selenium concentrations have been reported in various studies. Women should have optimal selenium levels when planning a pregnancy to optimize the growth and maturation of follicles [22].

Dietary selenium is obtained mainly from cereals, grains, meat, fish/seafood, eggs, dairy products, and nuts, which demonstrate a large variability in selenium content [17]. Selenium concentrations in plants and animal-based foods vary across the world according to the selenium content in and geochemical characteristics of the soil, the ability of consumable plants to take up selenium, and, finally, the diet of animals [23]. Cereal products account for ~50% of the daily intake of selenium, whereas ~35% of selenium demand is met by meat, fish, and poultry intake [24].

Given that dietary intake is the main source of both iodine and selenium, we aimed to assess the consumption of relevant products in women of reproductive age and the current iodine and selenium nutritional status in pregnant women in Latvia.

## 2. Materials and Methods

The consumption of iodine- and selenium-containing products by women of reproductive age (15–49) in the general population of Latvia was estimated from health survey data. The Health Behaviour Survey among the Latvian Adult Population is a population-based survey that has been repeated every two years since 1998 (from 1998 to 2008, a collaborative project of the Finbalt Health Monitor) [25,26]. In 2010, the method of data collection was changed from a postal survey with self-administered questionnaires to face-to-face interviews; therefore, we included surveys from 2010 to 2018 in our data analysis.

The representative sample from the general population aged 15–64 (surveys 2010–2014) or 15–74 (surveys 2016 and 2018) was selected using stratified random multistage sampling. Sample size estimates and stratification by age, sex, region and urbanization were based on the latest available population data provided by the Central Statistical Bureau of Latvia. After random selection of initial interview points, the random route procedure and quota method were used. For the purpose of our study, we restricted analysis of survey data to the reproductive age women.

Pregnant women not reporting any health problems were recruited at the Riga Maternity Hospital, which is the largest center for pregnancy care in Latvia. Overall, 129 women were enrolled in the first trimester of pregnancy during their first antenatal visit (weeks 6–8), and their serum selenium and serum selenium protein levels, iodine intake, thyroid function, and thyroid peroxidase antibodies (TPO-ab) levels, which are indicative of autoimmune processes, were measured. To determine dietary iodine sources, women were asked to complete the questionnaire concerning the use of iodine/selenium supplements and the consumption of seafood, dairy products and iodized household salt, as well as smoking history, previous thyroid diseases and parity.

The measurement of urinary iodine concentration (UIC) was based on the Sandell–Kolthoff reaction. In detail, 40 μL of urine samples were added to 160 μL of 1.0 M ammonium persulfate and heated for 60 min at 100 °C. The samples were cooled to room temperature, and 400 μL of arsenious acid solution (0.1 M in 0.5 M H_2_SO_4_) was added. The samples were mixed, and 200 μL of each sample was pipetted into a 96-well plate and allowed to stand for 15 min at room temperature. Then, 16 μL of ceric ammonium sulfate solution (0.15 M in 1.75 M H_2_SO_4_) was added to each sample and quickly mixed. Absorption was measured at 405 nm after a 30 min incubation at room temperature using a μQuant™ Microplate Spectrophotometer (BioTek, Winooski, USA). In parallel, standard iodine solutions (concentrations of 300, 200, 150, 100, 50, 20, and 0 μg/L) were treated identically. To clarify so-called matrix effects, methods used for iodine level measurement were validated by preparing standard solutions of iodine using different matrices (water or a random urine sample). The standard curves created using water and urine as the matrix were identical, indicating that the method was accurate and that the matrix effect on urinary iodine measurements was low. The urinary creatinine concentration was measured using the Jaffe method [27] so that the iodine concentration adjusted for the creatinine concentration (iodine/Cr) could be calculated. The creatine-standardized urinary iodine concentration (UIC) was calculated as a more reliable measure than random spot UIC measurement considering the great day-to-day variability in water intake [28].

Serum thyroid-stimulating hormone (TSH), serum-free thyroxine (fT4) and TPO-ab levels were measured by a chemiluminescence immunoassay (Siemens, Malvern, PA, USA), which was performed on an Advia Centaur XP (Siemens) analyzer by the E. Gulbis Laboratory (Riga, Latvia), which operates according to EN ISO 15189 standards.

In addition, through the EUthyroid project (a European Union-funded research project evaluating current national efforts aimed at preventing iodine deficiency disorders), urine and serum samples were reanalyzed at the central EUthyroid laboratory in Helsinki to compare and harmonize UIC and serum TSH, fT4, and TPO-ab values with those obtained by other EUthyroid partner countries throughout Europe.

The plasma selenium concentration was determined fluorometrically by using a “Cary Eclipse” fluorescence spectrophotometer (Varian, Inc., Houten, The Netherlands). Interlaboratory quality control was conducted by employing two standards—selenium AAS solution (Aldrich, St. Louis, MO, USA, Cat#24, 792-8) and Seronorm TE Serum Level I (Sero AS, Cat#201 405, Billingstad, Norway)—for the Seronorm™ Trace Elements-Controls Programme. External Quality Assessment Services were performed by Labquality Oy, Finland.

Selenoprotein P (SEPP) concentrations were measured using a Spark^®^ multimode microplate reader (Tecan Group Ltd., Mannedorf, Switzerland) by a validated commercial SELENOP-specific ELISA kit (Cusabio, Wuhan, China) for human cells and a Rat Selenoprotein P (Selenop) ELISA kit (Cusabio, Wuhan, China) according to the instructions of the supplier. Statistical analysis was performed using IBM SPSS Statistics for Windows, version 26.0. The results are expressed as means and standard deviations (SDs) or as medians and interquartile ranges (IQRs) and as the percentage of participants in the study. Two-sided Student’s *t*-test and the Mann–Whitney U test were used to compare differences between the two subgroups. Differences in prevalence estimates were tested using the χ^2^ test. Pearson’s or Spearman’s correlation coefficients were calculated depending on the normality of the data. P values below 0.05 were considered statistically significant.

The Ethical Committee of Riga Stradiņš University approved the study protocol on 4 September 2014. All participants signed the informed consent form. The Centre for Disease Prevention and Control is the owner of the fully anonymized population health survey data; the research datasets were prepared by the Centre staff and released to the researchers.

## 3. Results

### 3.1. Dietary Sources of Iodine and Selenium in Pregnant Women

The main sources of iodine and selenium in women of reproductive age in Latvia according to the population health survey and food questionnaire are presented in Figure 1. The use of iodized salt was the highest in 2010 (*p* < 0.001), when 13.4% of women reported that iodized salt was usually used at home; since then, less than 10% reported such behavior. Consumption of milk and dairy products as potential sources of iodine was substantially lower in 2018 than in previous survey years (*p* < 0.001); only 33.4% of women reported having at least two servings of milk or dairy products a day compared to 41–48% in earlier years. This decrease was observed in all age groups, see in Appendix A.

In addition, intake of bread as one of the main sources of selenium showed a significant decrease in recent years (*p* < 0.001): approximately 70% of reproductive age women reported consumption of more than three slices of any type of bread per day from 2010–2014, while only 54.7% and 45.5% of women reported it in 2016 and 2018, respectively. The same tendency was observed for all age groups (Appendix A).

Furthermore, the consumption of important sources of selenium and iodine such as meat and fish did not change substantially; approximately 70% of respondents reported eating either meat or meat products at least three days a week, and almost 60% reported eating some kind of fish at least once per week. However, in all surveys, fish intake was significantly lower among younger women (*p* < 0.001): only 39.4–48.2% of 15- to 24-year-old women had eaten fish at least once per week (Appendix A).

Table 1 presents the prevalence of the consumption of food items among both pregnant women in the study and women of reproductive age in the general population. The direct comparison of product intake was limited due to differences in the questionnaires. Nevertheless, in both populations, there was an association between the use of iodized salt and age: the older the women were, the greater the proportion who reported using iodized salt (*p* = 0.016 in pregnant women and *p* < 0.001 in the general population). Overall, the proportion of pregnant women who always used iodized salt was 8.5% (95% CI 4.8–14.5). This number does not differ from the 9.4% of regular users of iodized salt according to the Finbalt survey (the question to the general population was “What kind of salt do you usually use at home?”; “iodized salt” was one of the mutually exclusive answers).

On the other hand, a slight increase in the use of grain products and cereals was observed in 2018 compared to the previous survey years; 28.2% of women reported eating rice or pasta, and 33.8% reported eating porridge or breakfast cereal at least three days a week. In addition, in 2018, there was a substantial increase in the use of vitamins and supplements, which may have contributed to the intake of both iodine and selenium (*p* < 0.001). In the latest survey, 48.2% of reproductive age women reported the use of some vitamins and supplements in the last week, whereas ~30% did so in from 2010–2016. This increase was observed in all age groups.

Similarly, in both populations, there were age-dependent differences in fish and/or seafood consumption. Among pregnant women, younger women reported a lower intake of seafood, but the proportion of individuals who ate seafood at least once per week was the highest in the 35- to 40-year-old group at 68.8% (*p* = 0.049 according to age).

A high proportion (68.2%) of pregnant women were already using vitamins and supplements at their first antenatal visit; the percentage of users was particularly high among older pregnant women—more than 80%—but the difference age-related was not statistically significant. However, only 19 (14.7%) pregnant women used supplements containing sufficient amounts [29] of iodine (150 μg of iodine), and only 13 (10.1%) used supplements containing selenium (60 μg/day).

### 3.2. IODINE and Selenium Status in the 1st Trimester of Pregnancy

Among pregnant women at 6–8 weeks of gestation, the median UIC (IQR) was 147.2 (90.0–248.1) μg/gCr, and 52.8% of participants had a UIC below 150 μg/gCr (Figure 2). The mean serum selenium (SD) was 101.5 (35.6) μg/L, with 72.6% of participants having selenium levels below 120 μg/L and 30.1% having selenium levels below 80 μg/L, whereas the median selenoprotein P was 6.9 (3.1–9.0) mg/L.

Thyroid function was normal in all pregnant women. The mean TSH level was 1.1 (0.7) mIU/L, and the mean fT4 level was 14.2 (2.7) pmol/L, whereas 13.9% of women had a TPO-ab level above 60 IU/mL.

The differences in urinary iodine excretion, serum selenium concentration, selenoprotein *p* concentration and TPO-ab levels between subgroups were not statistically significant at the 0.05 level (Table 2). However, the number of dietary sources of iodine (iodized salt, milk and dairy products, seafood and vitamins/supplements, as listed and dichotomized in Table 2) showed a positive correlation with the UIC (Spearman’s rho = 0.217, *p* = 0.016).

There was no significant correlation between serum selenium and SEPP levels in our population (*p* = 0.636). Serum Se and SEPP levels did not differ between pregnant women with high levels of TPO-ab and those with normal levels (*p* = 0.248 and *p* = 0.938, respectively). No significant correlation was found between serum Se or SEPP and TSH levels (*p* = 0.225 and *p* = 0.532, respectively). In addition, no association was detected between serum Se or SEPP levels and age (*p* = 0.675 and *p* = 0.495, respectively).

There was no statistically significant correlation between any of the clinical and nutritional markers except TPO-ab levels, which were positively correlated with TSH levels (Spearman’s rho 0.28, *p* = 0.002).

## 4. Discussion

Considering the worrying and conflicting results from a previous study of pregnant Latvian women carried out in 2013 [30], campaigns to raise awareness in our population and guidelines for healthcare professionals were issued. To assess whether these efforts have resulted in improved iodine and selenium nutrition, we assessed the consumption of relevant products in women of reproductive age and the current iodine and selenium nutritional status of pregnant women in Latvia. The analysis of the survey data from women of reproductive age showed that in the period from 2010 to 2018, the consumption of several iodine- or selenium-rich products, such as dairy products and bread, decreased. Only 1/3 of women in 2018 reported consuming at least two servings of milk or dairy products a day (compared to 40% or more in previous surveys). The consumption of bread has decreased 1.5 times (the percentage of individuals who reported consuming at least four slices of any bread daily decreased from 70% to 45.5%). Since the 2012 survey, less than 10% of women have reported using iodized salt. On the other hand, there was an increase in the consumption of grain products (rice, pasta, porridge, and breakfast cereal) observed in 2018. However, the most striking increase was in the use of vitamins and supplements: almost half (48.2%) of women of reproductive age in Latvia reported using supplements in the 2018 survey. In pregnant women, the proportion using vitamins and supplements was even higher—68.2%. However, only 10–15% used supplements containing sufficient amounts of iodine or selenium.

The results of the present study showed that approximately half of pregnant women reach adequate iodine supply in the first trimester of pregnancy despite no obvious changes in dietary habits in past years. One possible explanation for improved iodine nutrition might be increased consumption of imported foods from countries where salt fortification with iodine is mandatory and iodized salt is used in industrial food production. The use of iodized salt in the manufacturing of frequently consumed processed foods may have a more considerable impact on the daily iodine intake of consumers than the use of iodized salt in the household. The survey data for young (18- to 35-year-old) adults in Latvia [31] revealed that grain products (bread, breakfast cereals, biscuits, and cakes) and meat and meat products were the main sources of salt intake (each group accounted for 29.5% of the daily intake). Furthermore, the import of these products has increased over time [32], and in 2017, the dominant country from which products were imported was Poland, where a mandatory iodine deficiency prevention program had been introduced; all meat and edible meat offal products imported from European countries (in Euro), 21.4% were from Poland. In addition, the second largest importer of meat and edible meat offal products in 2017 was Denmark (11.2% of total meat imports, Euro), where salt fortification with iodine is also mandatory. Poland was also the leader in importing prepared cereals, flour, starch, milk or pastry products in 2017 (14.8% of the total import of these products, Euro).

An important source (30–60%) of iodine is milk and milk products [33]. The average iodine concentration in milk samples in Latvia (457.6 μg/L) [34] is higher than that in other countries because of iodine-rich cattle feed supplement use and iodine-containing disinfectants used by farms. In addition, iodine fortification on large farms has increased during the last three years, and locally produced dairy products are preferred by the population in Latvia. In addition, more common use of supplements, even those with relatively low iodine content, may lead to better iodine status in pregnant women in Latvia.

In our previous analysis of thyroid medication use in Latvia during 2011–2014, an increase in the prevalence of thyroid autoimmune diseases in all age groups was noted [35]. Recent recommendations for women at risk of postpartum thyroid deficiency suggest a selenium intake of 200 μg/day or higher [36] to reduce excess hydrogen peroxide and autoimmunity. Serum selenium levels in the 1st trimester (101.5 μg/L) were lower in our population than in populations of some countries, such as the United States (151 μg/L), Japan (140.2 ± 12.4 μg/L), Nigeria (107.4 ± 15.8 μg/L), and Finland (106 ± 15 μg/L), but higher than values reported in other countries, such as Germany (89 ± 1 μg/L), Serbia (63 μg/L), Finland (59 μg/L), and Poland (53.4 ± 8.0 μg/L). Approximately one-third (30.1%) of pregnant women in Latvia had selenium levels below 80 μg/L, which corresponds to a status of selenium deficiency [37].

Selenium intake levels vary among different regions and largely depend on the selenium content of foods. The main sources of selenium for the Latvian population are bread, cereals, meat, and fish. In 2011, dietary selenium intake was assessed in 990 pregnant women in Latvia using 24-h dietary recalls on two consecutive days (FC_PREGNANTWOMEN 2011). The average selenium intake was 50.3 μg/day, which is below the recommended daily intake of 60 µg/day for pregnant women according to European authorities [38] and below the new value of adequate intake of 70 µg/day for women in pregnancy set by the European Food Safety Authority in 2014 [17]. However, daily dose recommendations for selenium vary from one country to another.

Selenium may reduce thyroid inflammatory activity and the development of postpartum thyroiditis and permanent hypothyroidism in patients with autoimmune thyroiditis [36,39]. Negro et al. reported that TPO-ab-positive pregnant women treated with selenium 200 μg per day during pregnancy and postpartum had a significantly lower incidence of postpartum thyroiditis and permanent hypothyroidism compared to those in the untreated group [36]. Although currently there is a lack of sufficient evidence to recommend routine selenium supplementation during pregnancy in TPO-ab–positive women [40], further randomized studies should be conducted on this subject.

The selenium status in the Latvian general population has not been assessed, but this study reported a mean serum selenium level of 101.5 μg/L in pregnant women in Latvia (~30% being deficient, below 80 μg/L). The optimal selenium status and the best biomarker of selenium sufficiency in pregnancy have not been clearly defined by international experts. The SEPP concentration reflects selenium resources in the organism and appears to be a better marker of selenium status than the concentration of plasma selenium, which also includes selenomethionine incorporated in the protein structure [12]. Although several commercial kits for the quantification of SEPP in the plasma are available, reference values for SEPP are not currently established as there are significant variations in the serum concentrations of SEPP obtained by different techniques, immunoassays, and laboratories [41]. In the current study, the median SEPP level was 6.9 (3.1–9.0) mg/L. In comparison, the median SEPP level in controls (n = 993) in the study nested within a Danish cohort was 5.5 (3.5–8.0) mg/L [42]. Concentrations of the SEPP were measured in 966 healthy individuals participating in the large European Prospective Investigation into Cancer and Nutrition (EPIC) study and the mean SEPP level was 4.4 mg/L [43]. A reference range of 2.56–6.63 mg/L for serum SEPP concentration has recently been proposed based on the data obtained from EPIC study [44]. However, data on SEPP concentrations in pregnant women are scarce.

Butler and Whanger reported that during pregnancy, 13–17% of selenium is incorporated into plasma GPX, 50–60% is incorporated into SEPP, and 23–32% is bound to albumins [45].

We did not find any significant correlation between serum selenium and SEPP levels in early pregnancy, indicating that either the selenium status of pregnant women was at the level needed for full expression of selenoproteins or, more likely, that increasing estrogene levels increase SEPP levels, similar to thyroxine-binding globulin and cortisol-binding globulin [46]. In addition, animal experiments have provided evidence that the thyroid and the brain specifically take up selenium in cases of selenium deficiency [47]. Therefore, more robust evidence of the efficacy of SEPP levels as a biomarker of selenium status in pregnancy is needed.

The limitation of our data is that health survey including consumption of iodine- and selenium-containing products by women of reproductive age in the general population was estimated from health survey covered all the country, but the survey in pregnant women, tested on biochemical parameters (serum selenium and serum selenium protein levels, iodine intake, thyroid function, and TPO-ab levels) was limited to a capital maternity hospital.

## 5. Conclusions

In conclusion, iodine nutrition in the Latvian population of pregnant women was between insufficient and adequate. This result was accompanied by borderline selenium deficiency, and 30% of women were selenium deficient. Supplements were frequently used, but most did not contain sufficient amounts of iodine and selenium to correct the inadequate intake through food. Considering the trend of lower consumption of some iodine- and selenium-rich products but more common use of supplements, pregnant women and women planning a pregnancy should be educated in paying attention to the contents of supplements.

## Figures and Tables

**Figure 1 medicina-57-01211-f001:**
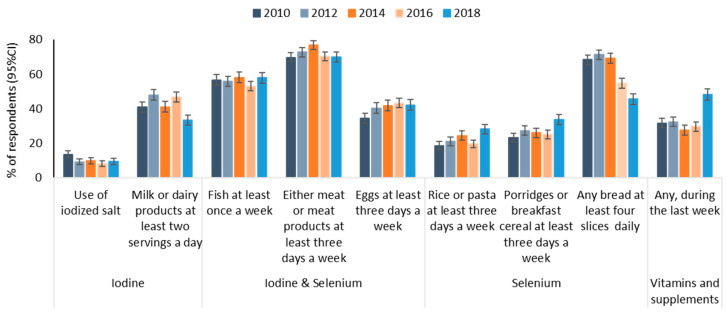
The main sources of iodine and selenium in women of reproductive age according to population-based health surveys in Latvia.

**Figure 2 medicina-57-01211-f002:**
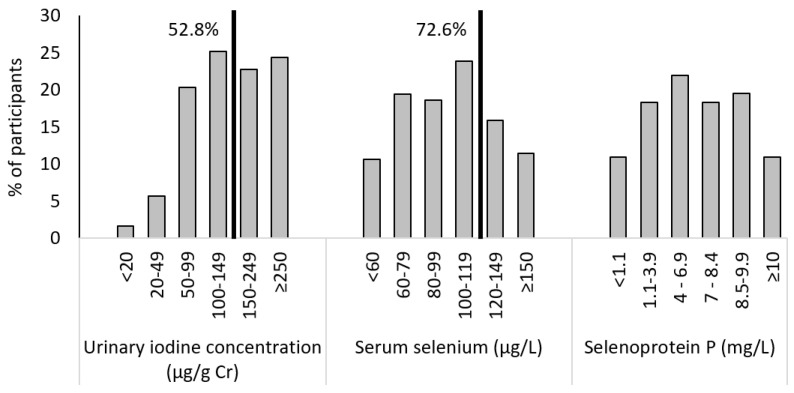
Frequency distribution of urinary iodine concentrations, serum selenium concentrations and selenoprotein *p* concentrations in pregnant women in Latvia at the first antenatal visit (weeks 6–8).

**Table 1 medicina-57-01211-t001:** The use of iodine-rich products by age group according to pregnant women’s questionnaire responses and the 2018 population-based health survey.

	*n*	Proportion of Study Participants, % (95% CI)			
** *Survey of pregnant women* **					
		Use of iodized salt always or sometimes	Consumption of at least two servings of milk or dairy products a day	Consumption of seafood at least once a week	Use of any vitamins and supplements during the last 3 weeks
Total	129	37.2 (29.4–45.8)	31.0 (23.7–39.4)	48.1 (39.6–56.6)	68.2 (59.8–75.6)
Age, years					
17–24	20	15.0 (5.2–36.0)	15.0 (5.2–36.0)	30.0 (14.6–51.9)	65.0 (43.3–81.9)
25–29	51	37.3 (25.3–51.0)	33.3 (22.0–47.0)	49.0 (35.9–62.3)	66.7 (53.0–78.0)
30–34	42	40.5 (27.0–55.5)	31.0 (19.1–46.0)	47.6 (33.4–62.3)	66.7 (51.6–79.0)
35–40	16	56.3 (33.2–76.9)	43.8 (23.1–66.8)	68.8 (44.4–85.8)	81.3 (57.0–93.4)
P for trend (χ^2^ test)		0.016	0.124	0.049	0.381
P (χ^2^ test)		0.075	0.291	0.146	0.693
** *Women of reproductive age in Latvia: The Health Behaviour survey among the Latvian Adult Population, 2018* **					
		Use of iodized salt	Consumption of at least two servings of milk or dairy products a day	Consumption of fish at least once a week	Use of any vitamins and supplements during the last week
Total	991	9.4 (7.7–11.4)	33.4 (30.5–36.4)	57.9 (54.8–61.0)	48.2 (45.1–51.4)
Age, years					
15–24	247	3.2 (1.6–6.3)	30.4 (25.0–36.4)	48.2 (42.0–54.4)	38.9 (33.0–45.1)
25–29	138	6.5 (3.5–11.9)	34.1 (26.7–42.3)	63.8 (55.5–71.3)	54.4 (46.0–62.4)
30–34	152	17.8 (12.5–24.6)	32.2 (25.3–40.0)	59.9 (51.9–67.3)	49.3 (41.5–57.2)
35–49	454	10.8 (8.3–14.0)	35.2 (31.0–39.7)	60.8 (56.2–65.2)	51.1 (46.5–55.7)
P for trend (χ2 test)		<0.001	0.227	0.006	0.014
P (χ2 test)		<0.001	0.607	0.004	0.001

**Table 2 medicina-57-01211-t002:** Iodine and selenium status in pregnant women in association with demographic and dietary factors.

	*N*	Urinary Iodine Concentration, μg/g Cr		Serum Selenium Concentration, μg/L		Selenoprotein P Concentration, mg/L	TPO-ab Level
		Median (IQR)	<150 μg/g Cr, %	Mean (SD)	<120 μg/L, %	Median (IQR)	>60 IU/mL, %
** *Age, years* **							
17–24	20	128.2 (78.5–184.5)	57.9	96.4 (27.6)	77.8	8.4 (3.8–9.5)	10.0
25–29	51	147.2 (114.0–281.7)	55.1	103.0 (40.1)	73.5	6.0 (1.4–8.8)	9.8
30–34	42	134.5 (80.6–272.1)	52.5	98.8 (32.9)	67.6	7.2 (4.4–9.2)	18.4
35–40	16	171.0 (106.8–229.0)	40.0	103.2 (25.9)	81.8	4.4 (3.0–6.0)	23.1
** *First pregnancy* **							
Yes	60	154.9 (110.5–248.1)	49.2	101.6 (33.0)	73.6	6.9 (1.8–9.1)	10.2
No	69	134.0 (88.2–263.4)	56.3	99.8 (36.4)	72.9	7.0 (3.7–8.8)	17.5
** *Use of iodized salt* **							
Always or sometimes	48	153.8 (110.8–293.1)	44.4	97.3 (29.8)	82.1	6.8 (1.5–8.9)	19.6
No	81	134.0 (82.5–201.0)	57.7	102.5 (37.1)	68.5	7.1 (3.4–9.0)	10.5
** *Consumption of milk or dairy products* **							
At least two servings a day	40	149.0 (114.0–330.4)	51.4	101.2 (35.1)	75.7	5.9 (2.0–8.9)	22.5
Less than two servings a day	89	145.2 (82.5–195.4)	53.5	100.4 (34.7)	72.0	7.1 (3.2–9.0)	9.8
** *Consumption of seafood* **							
At least once a week	62	147.2 (102.8–265.3)	52.5	97.0 (36.5)	75.0	5.8 (2.5–9.3)	14.0
Less than once a week	67	146.0 (89.5–195.8)	53.1	103.8 (33.0)	71.7	7.1 (3.5–8.9)	13.8
** *Use of any vitamins and supplements during the last 3 weeks* **							
Yes	88	154.4 (105.8–238.1)	47.7	97.5 (35.8)	76.3	6.0 (1.7–9.0)	14.8
No	41	124.5 (87.2–250.2)	64.9	109.7 (34.1)	64.9	7.8 (4.4–9.0)	12.2
*Total*	129	147.2 (90.0–248.1)	52.8	101.5 (35.6)	72.6	6.9 (3.1–9.0)	13.9

## Data Availability

We exclude this statement as we did not report any date.

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
