# Peer review of "Assessment of Iodine and Selenium Nutritional Status in Women of Reproductive Age in Latvia"

_medicina, 2021, doi:10.3390/medicina57111211_

Round 1
Reviewer 1 Report
Authors have addressed and revised the manuscriptaccording to my former comments.
The paper can be published in this form.
Author Response
Thank you indeed for your revisions!
Reviewer 2 Report
One remark: the epidemiological nutritional survey covered all the country; biochemical nutritional survey in pregnant women was limited to a capital maternity hospital. This should be emphasized as a "limit" in the Discussion section of your your manuscript.
Editing correction:
line 168: "supplierStatistical" to be changed in "supplier. Statistical"
Author Response
Thank you for your revisions and remark!
Point 1: The following text is added at the end of discussion [lines 370-375] according to your suggestions:
"The limitation of our data is that health survey including consumption of iodine- and selenium-containing products by women of reproductive age in the general population was estimated from health survey covered all the country, but the survey in pregnant women, tested on biochemical parameters (serum selenium and serum selenium protein levels, iodine intake, thyroid function, and TPO-ab levels) was limited to a capital maternity hospital."
Point 2: line 168: "supplierStatistical" is changed in "supplier. Statistical